

# The impact of sea-level rise on tidal characteristics around Australasia

Alexander Harker[1,2], J.A. Mattias Green[2], and Michael Schindelegger[1]

[1]Institute of Geodesy and Geoinformation, University of Bonn, Bonn, Germany
[2]School of Ocean Sciences, Bangor University, Menai Bridge, United Kingdom

**Correspondence:** Alexander Harker (harker@igg.uni-bonn.de)

**Abstract.** An established tidal model, validated for present-day conditions, is used to investigate the effect of large levels of sea-level rise (SLR) on tidal characteristics around Australasia. SLR is implemented through a uniform depth increase across the model domain, with a comparison between the coastal boundary being treated as impenetrable or allowing low-lying land to flood. The complex spatial response of the semi-diurnal constituents, $M_2$ and $S_2$, is broadly similar, with the magnitude of $M_2$'s response being greater. The most predominant features of this response are large amplitude changes in the Arafura Sea and within embayments along Australia's north-west coast, and the generation of new amphidromic systems within the Gulf of Carpentaria and south of Papua, once water depth across the domain is increased by 3 and 7 m respectively. Dissipation from $M_2$ increases around the islands in the north of the Sahul shelf region and around coastal features along north Australia, leading to a notable drop in dissipation along Eighty Mile Beach. The diurnal constituent, $K_1$, is found to be amplified within the Gulf of Carpentaria, indicating a possible change of resonance properties of the gulf. Coastal flooding has a profound impact on the response of tidal amplitudes to SLR, particularly $K_1$, by creating local regions of increased tidal dissipation and altering the shape of coastlines.

## 1 Introduction

The behavior of the ocean is of great consequence to human life. At sea level the land is often flat and well-suited for farming, whilst coastal waters can be used for transport, trade, and as a source of food. This makes coastal areas an attractive location for human populations to settle (e.g., Pugh and Woodworth, 2014). Being a large island country, 85% of the population of Australia live within 50 kilometers of the ocean, and the recreation and tourism industries located along the coast are a key part of Australia's economy (Watson, 2011). As such, Australia is particularly sensitive to both short term fluctuations in sea level (e.g. tidal and meteorological effects) and long term changes in mean sea level. The combination of these effects upon local extreme water levels must be considered when predicting which areas may be at risk of flooding (Muis et al., 2016). With sea levels around Australia expected to rise (Muis et al., 2016; McInnes et al., 2015; Zhang et al., 2017), there are serious implications for urban planning and coastal protection strategies in low-lying areas (Holleman and Stacey, 2014; Woodworth, 2017).





Peaks in spring-neap or lunar-nodal tidal cycles, combined with severe weather events, are often one of the key components of extreme water levels (Haigh et al., 2011), therefore an understanding of how tidal ranges are expected to change is crucial to understanding the risk upon coastal communities and infrastructure. Being the dynamical response of the oceans to gravitational forcing, tides are sensitive to a variety of parameters, including the amount of water contained within the ocean basins. Such

changes in water depth may have an impact on the speed at which the tide propagates and the dissipation of tidal energy, and it may change the resonant properties of an ocean basin. As an extreme example, during the last glacial maximum, when sea level was approximately 120 m lower than present day, the tidal amplitude of the $M_2$ constituent in the North Atlantic was greater by a factor of two or more because of amplified tidal resonances there (Egbert et al., 2004; Wilmes and Green, 2014). The tides also are a major influencing factor on coastal navigation, ecology, sedimentation and erosion (Mawdsley et al.,

2015). Consequently, understanding the oceans' response to tidal forces under changing sea-level has been a subject of recent research, at both regional (Greenberg et al., 2012; Pelling et al., 2013; Pelling and Green, 2013; Carless et al., 2016) and global scales (Müller et al., 2011; Pickering et al., 2012; Wilmes et al., 2017; Schindelegger et al., 2018).

Current estimates of the global average change in sea level over the last century are a rise of between 1.2–1.7mm yr$^{-1}$, (Church et al., 2013; Hay et al., 2015; Dangendorf et al., 2017). There are, however, significant inter-annual and decadal-scale

fluctuations, and in the past decade global sea-level rise (SLR) has been measured as high as 3.2 mm yr$^{-1}$ (Church and White, 2011). Sea-level change is neither temporally nor spatially uniform as a multitude of physical processes contribute to regional variations across the globe (Cazenave and Llovel, 2010; Slangen et al., 2012). Studies suggest that global sea level may rise by up to 1 m by the end of the 21$^{st}$ century, and by up to 3.5 m by the end of the 22$^{nd}$ century (Vellinga et al., 2009; DeConto and Pollard, 2016). Some of this signal is attributed to an increasing ocean heat content causing thermal expansion of the water

column, but most of the rise and acceleration in sea level are due to enhanced melting of the Antarctic and Greenland Ice Sheets (Church et al., 2013; Rietbroek et al., 2016). The effect of vertical land motion on relative sea level should also come under consideration; however around the Australian coastline the effect is small (White et al., 2014) so for the following this phenomena is neglected.

Here, we expand previous tides and SLR investigations to the area surrounding Australasia and the Sahul shelf, which has

received little attention so far despite the north-west Australian shelf alone being responsible for a large amount of energy dissipation on par with the Yellow sea or the Patagonian shelf (Egbert et al., 2004). We specifically study the region's tidal response to a uniform SLR signal, as well as implementing both inundation of land or coastal flood-defenses. This area is a tidally active region, with areas currently experiencing tidal resonances, e.g. the Gulf of Carpentaria (Webb, 2012). It is therefore expected that we will see large differences in the tidal signals with even moderate SLR, as it is known that a (near-)

resonant tidal basin is highly sensitive to bathymetric changes (e.g., Green, 2010). In what follows we introduce OTIS, the dedicated tidal modeling software used, and the simulations performed (Sections 2.1–2.3). To ground our considerations of future tides on a firm observational basis, we conduct extensive comparisons to tide gauge data in Section 2.4. Section 3 presents the results, and the paper concludes in the last section with a discussion.





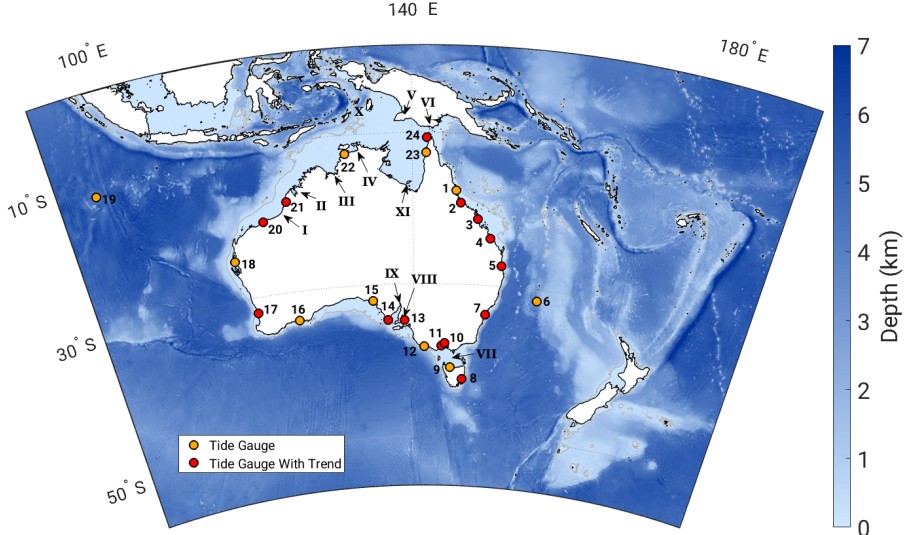

**Figure 1.** Bathymetry of the model domain and tide gauge sites used in the analysis (colored dots): cf Table 1. Locations at which $M_2$ trends were estimated are shown in red. Regions mentioned in subsequent sections of the paper are marked on the map:I - Eighty Mile Beach, II - King Sound, III - Joseph Bonaparte Gulf, IV - Van Dieman Gulf, V - Yos Sudarso Island, VI - Torres Strait, VII - Bass Strait, VIII - Gulf St. Vincent, IX - Spencer Gulf, X - Arafura Islands, XI - Wellesley Islands.

## 2 Modeling future tides

### 2.1 Model configuration

We use OTIS—the Oregon State University Tidal Inversion Software—to simulate the effects of SLR on the tides around Australia. OTIS is a portable, dedicated, numerical shallow water tidal model which has been used extensively for both global

5  and regional modeling of past, present and future ocean tides (e.g., Egbert et al., 2004; Pelling and Green, 2013; Wilmes and Green, 2014; Green et al., 2017). It is highly accurate both in the open ocean and in coastal regions (Stammer et al., 2014), and it is computationally efficient. The model solves the linearized shallow-water equations (e.g., Hendershott, 1977) given by

$$\frac{\partial \mathbf{U}}{\partial t} + \mathbf{f} \times \mathbf{U} = -gH\nabla(\zeta - \zeta_{EQ} - \zeta_{SAL}) - \mathbf{F} \tag{1}$$

$$\frac{\partial \zeta}{\partial t} = -\nabla \cdot \mathbf{U} \tag{2}$$

10  where $\mathbf{U}$ is the depth integrated volume transport, which is calculated as tidal current velocity $\mathbf{u}$ times water depth $H$. $\mathbf{f}$ is the Coriolis vector, $g$ denotes the gravitational constant, $\zeta$ stands for tidal elevation with respect to the moving seabed, $\zeta_{SAL}$ denotes the tidal elevation due to ocean self-attraction and loading (SAL), and $\zeta_{EQ}$ is the equilibrium tidal elevation. $\mathbf{F}$ represents energy losses due to bed friction and barotropic-baroclinic conversion at steep topography. The former is represented



**Table 1.** Start and end dates of the analyzed tide gauge records, including names and running index for identification in Figure 1

| Station ID | Name | Time Span | Source |
|---:|---|---|---|
| 1 | Mourilyan Harbour | 1986–2014 | GESLA |
| 2 | Townsville | 1980–2014 | GESLA |
| 3 | Hay Point | 1985–2014 | UHSLC |
| 4 | Gladstone | 1982–2014 | UHSLC |
| 5 | Brisbane | 1985–2016 | GESLA |
| 6 | Lord Howe Island | 1992–2014 | GESLA |
| 7 | Fort Denison | 1965–2017 | GESLA |
| 8 | Spring Bay | 1986–2017 | GESLA |
| 9 | Burnie | 1985–2014 | GESLA |
| 10 | Williamstown | 1976–2014 | UHSLC |
| 11 | Geelong | 1976–2014 | UHSLC |
| 12 | Portland | 1982–2014 | GESLA |
| 13 | Port Adelaide | 1976–2014 | GESLA |
| 14 | Port Lincoln | 1967–2014 | UHSLC |
| 15 | Thevenard | 1966–2014 | GESLA |
| 16 | Esperance | 1985–2017 | GESLA |
| 17 | Fremantle | 1970–2014 | GESLA |
| 18 | Canarvon | 1991–2014 | GESLA |
| 19 | Cocos Islands | 1991–2017 | GESLA |
| 20 | Port Hedland | 1985–2014 | UHSLC |
| 21 | Broome | 1989–2017 | UHSLC |
| 22 | Darwin | 1991–2017 | GESLA |
| 23 | Weipa | 1986–2014 | GESLA |
| 24 | Booby Island | 1990–2017 | UHSLC |

by the standard quadratic law:

$$\mathbf{F}_B = C_d \mathbf{u}|\mathbf{u}| \tag{3}$$

where $C_d = 0.003$ is a non-dimensional drag coefficient, and $\mathbf{u}$ is the total velocity vector for all the tidal constituents. The parameterization for internal tide drag, $\mathbf{F}_w = C|\mathbf{U}|$, includes a conversion coefficient $C$, which is defined as (Zaron and Egbert, 2006; Green and Huber, 2013)

$$C(x,y) = \gamma \frac{(\nabla H)^2 N_b \bar{N}}{8\pi^2 \omega} \tag{4}$$

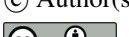



Here, $\gamma = 50$ is a scaling factor, $N_b$ is the buoyancy frequency evaluated at the sea bed, $\bar{N}$ is the vertical average of the buoyancy frequency, and $\omega$ is the frequency of the tidal constituent under evaluation. Values of both $N_b$ and $\bar{N}$ follow from the prescription of horizontally uniform stratification $N(z) = N_0 \exp(-z/1300)$, where $N_0 = 5.24 \times 10^{-3}\,\mathrm{s}^{-1}$ has been obtained from a least-squares fit to present-day climatological hydrography (Zaron and Egbert, 2006).

The model solves equations (1)–(2) using forcing from the astronomic tide generating potential only (represented by $\zeta_{EQ}$ in Eq. (1)). An initial spin-up from rest of over 7 days is followed by a further 15 days of simulation time, on which harmonic analysis is performed to obtain the tidal elevations and transports. Here, we focus our investigation on the $M_2$, $S_2$ and $K_1$ constituents only. The model bathymetry comes from the ETOPO1 dataset (Amante and Eakins, 2009, see Fig. 1 for the present domain), which was averaged to ⅟₂₀° horizontal resolution. At the open boundaries model heights were constrained to

elevation data from TPXO8 (Egbert and Erofeeva, 2002, updated version), and the TPXO8 data was also used to validate the model, alongside tide gauge data at 24 locations; see Section 2.4.

## 2.2 Dissipation computations

The computation of tidal dissipation rates, $D$, was done following Egbert and Ray (2001):

$$D = W - \nabla \cdot P \tag{5}$$

Here, $W$ is the work done by the tide-generating force and $P$ is the energy flux given by

$$
\begin{aligned}
W &= g\rho \langle \mathbf{U} \cdot \nabla(\eta_{SAL} + \eta_{EQ}) \rangle & (6)\\
P &= g \langle \eta \mathbf{U} \rangle & (7)
\end{aligned}
$$

where the angular brackets mark time-averages over a tidal period.

## 2.3 Implementing Sea-Level Rise

The model runs are split into two sets. In the first, we allow new low-lying grid cells to flood as sea level rises, whereas in the second set we introduce vertical walls at the present day coastline. Following Pelling et al. (2013) we denote these sets "flood" (FL) and "no flood" (NFL), respectively. A range of SLR scenarios are investigated in both sets via the implementation of a uniform depth increase across the entire domain of 1, 3, 5, 7, 10, 15 and 20 m. Additional runs were conducted with future changes in water depth extrapolated from sea surface trend patterns as observed by satellite altimetry (cf. Carless et al., 2016;

Schindelegger et al., 2018). While such projections contain not only the actual long term trend of sea level but also significant (sub-)decadal variability, little difference was found for tidal perturbations with respect to our uniform SLR scenarios. Hence in the following, the focus is on the latter. We choose 1 m and 7 m SLR because they best exemplify the changes to tidal characteristics across the domain and also correspond to a high but probable level for the end of this century and an extreme case in which large levels of ice sheet collapse has occurred (e.g., Wilmes et al., 2017).



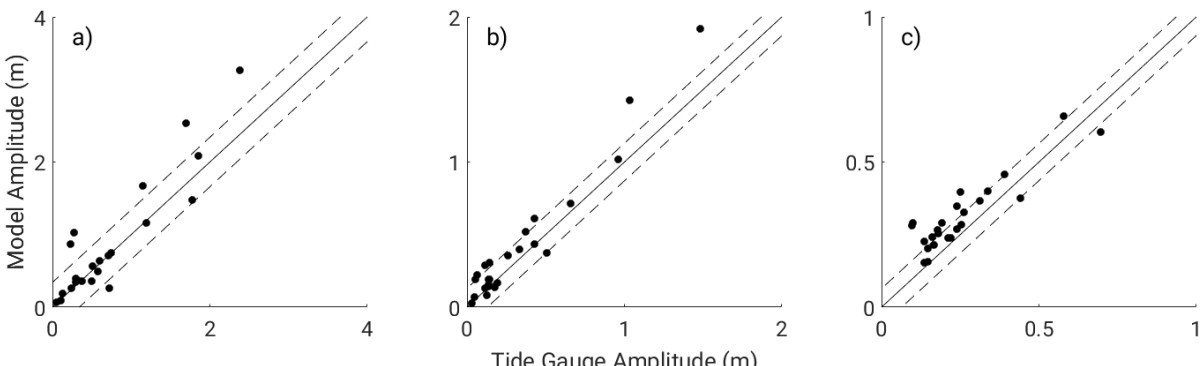

**Figure 2.** The constituent amplitude at each tide gauge position calculated from GESLA and UHSLC data and the model control simulation for a) $M_2$, b) $S_2$, and c) $K_1$. The solid line marks zero difference line. The dashed lines mark one standard deviation of the difference.

## 2.4 Model Validation

For validation of our numerical experiments, time series of hourly sea level data from 24 stations around Australasia were obtained from the Global Extreme Sea Level Analysis Version 2 (GESLA-2; Woodworth et al., 2017) and the University of Hawaii Sea Level Center (UHSLC; Caldwell et al., 2015); see Table 1. Record lengths are short with a few exceptions, but

all time series span at least 28 years to allow for an appropriate representation of the 18.61-year nodal cycle in lunar tidal constituents (Haigh et al., 2011). Upon removal of years missing more than 25% of hourly observations, a three-tiered least-squares fitting procedure was applied to (i) extract mean $M_2$, $S_2$, and $K_1$ tidal constants of amplitude $H$ and phase lag $G$, (ii) deduce linear $M_2$ trends in $H$ and $G$ over the complete time series at each station, and (iii) estimate the corresponding long-term trend in mean sea level (MSL). 10 out of 24 tide gauge stations yielded statistically insignificant $M_2$ amplitude trends at

the 95% confidence level and were thus excluded from the model trend validation below; see Fig. 1 for a graphical illustration.

     The processing protocol, essentially taken from Schindelegger et al. (2018), rests upon a separation of tidal and non-tidal residuals from the longer-term MSL component through application of a 4-day moving average with Gaussian weighting. High-frequency filter residuals obtained thereof were harmonically analyzed for 68 tidal constituents using the Matlab® UTide software package (Codiga, 2011), with analysis windows either set to the entire time series (step i) or shifted on an annual

basis (step ii). In both cases, we configured UTide for standard least squares and a white-noise approach in the computation of confidence intervals. Subsequent regressions of annual $M_2$ tidal constants were performed with a functional model composed of a linear trend, a lag one-year autocorrelation, AR(1), and sinusoids to account for nodal modulations. Trends in MSL were likewise determined through regression under AR(1) assumptions, upon a priori reduction of the influence of the 18.61-year equilibrium tide.

A comparison of the constituents amplitudes calculated from the OTIS control simulation for the present-day bathymetry to the $M_2$, $S_2$, and $K_1$ amplitudes from harmonic analysis of the tide gauge data (see Fig. 2) gives a root mean square (RMS) error of 36 cm for $M_2$, 15 cm for $S_2$, and 9 cm for $K_1$. Excluding those sites where the difference in amplitude between the





control simulation and tide gauge data is beyond $\pm 2$ standard deviations lowers the error to 21 cm for $M_2$, 9 cm for $S_2$ and 6 cm for $K_1$. The amplitude for $K_1$ is overestimated in the model since the internal tide drag coefficient is evaluated at the $M_2$ frequency. These statistics give confidence in the model's accuracy and in the assertion that areas where the disagreement between model and tide gauge data is largest are probably where the resolution of the model has not resolved small island or

near-shore features. Additional comparisons with gridded $M_2$ data were performed using the TPXO8 inverse solution, linearly interpolated to the grid of the model domain (see http://volkov.oce.orst.edu/tides for details). The RMS difference between the model and the TPXO8 data was less than 9 cm for $M_2$, and 4 cm for both $S_2$ and $K_1$. The variance capture (VC) of the control was also calculated to see how well the overall character of the tidal constituents was represented (e.g., Pelling and Green, 2013):

$$\text{VC} = 100 \left[ 1 - \left( \frac{\text{RMSD}}{\text{S}} \right)^2 \right] \tag{8}$$

where RMSD is the RMS difference between the control simulation and TPXO8, and S denotes the RMS standard deviation of the TPXO8 amplitudes. The VC was above 90% for all constituents, and above 95% for $M_2$.

For validation of the simulated $M_2$ changes under SLR, we followed Schindelegger et al. (2018) and condensed measured $M_2$ trends $(\partial H, \partial G)$ and MSL rates $(\partial s)$ to response coefficients in amplitude $(r_H = \partial H / \partial s)$ and phase $(r_G = \partial G / \partial s)$.

Simulated amplitude and phase changes from the 1 m NFL run at the location of 14 tide gauges were interpolated from nearest neighbor cells and also converted to ratios of $r_H$ and $r_G$. Graphical comparisons in Fig. 3 indicate that the model captures the sign of the observed $M_2$ amplitude response in 10 out of 14 cases and reproduces much of the in situ variability at approximately half of the analyzed stations (e.g., Booby Island, Williamstown, Geelong, Port Lincoln, Port Hedland, Broome). Model-to-data disparities on the northeastern seaboard (Hay Point, Gladstone) are markedly reduced in comparison to Schindelegger et al.

(2018, their Fig. 7), presumably due to the higher horizontal resolution of our setup in a region of ragged coastline features. Neither the increase of $M_2$ amplitudes at Townsville nor the pronounced reduction of the tide at Fort Denison (Sydney harbor) can be explained by SLR perturbations in the tidal model; both signals may be the effect of periodic dredging to maintain acceptable water depths for port operations; cf. Devlin et al. (2014) for a similar analysis. The decrease of the small $M_2$ tide at Spring Bay, Tasmania, (20 cm m$^{-1}$ of SLR) remains puzzling though, given that the gauge is open to the sea and unaffected by

harbor activities or variable river discharge rates. Mechanisms other than SLR, such as modulation of the internal tide due to stratification changes along its path of propagation (Colosi and Munk, 2006), need to be thoroughly addressed to complete the picture of secular changes in the surface tide. Despite these limitations, our 1 m NFL simulation captures good portions of the patterns of $M_2$ amplitude changes seen in tide gauge records around Australasia. Moreover, on time scales of centuries, water column depth changes due to SLR will outweigh tidal perturbations from other physical mechanisms. Hence, we conclude that

our simulations lend themselves well for analysis of probable future tidal changes over a wider area.



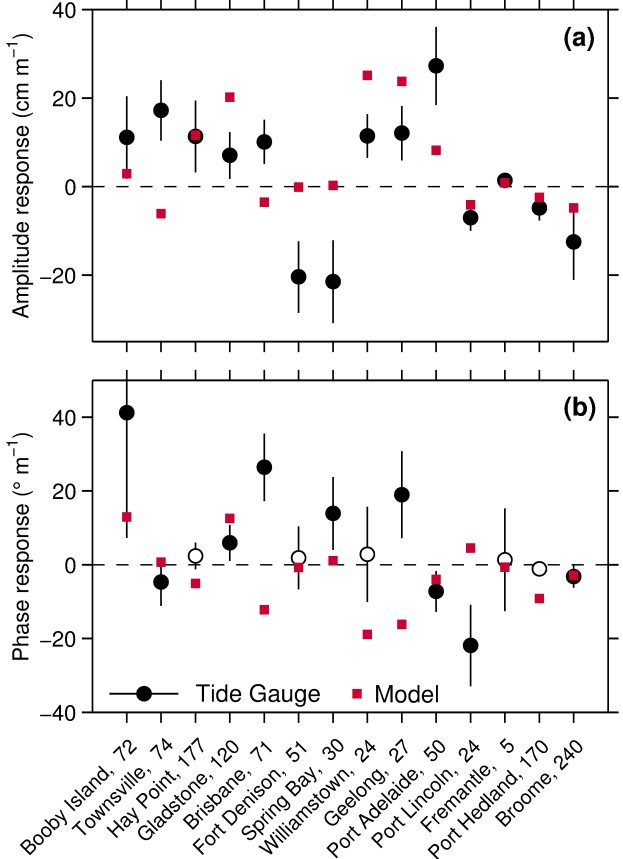

**Figure 3.** Observed and modeled $M_2$ response coefficients in (a) amplitude $H$ and (b) phase lag $G$ per meter of SLR. Model values (red squares) are based on the 1-m NFL simulations, while tide gauge estimates at 14 out of 24 locations are shown in black. Error bars correspond to two standard deviations, propagated from the trend analyses of sea level and annual tidal estimates. Stations with insignificant phase trends (at the 95% confidence level) are shown as white markers in panel (b). Numbers at the end of the station labeling indicate mean observed $M_2$ amplitudes (cm)

## 3 Results

In general, for all constituents, the amplitude difference with respect to present day is largely proportional to the imposed level of SLR (cf. Idier et al., 2017). No constituent saw significant differences between the FL and NFL scenarios for 1 m SLR, likely due to the fact that allowing land to flood at a 1 m SLR scenario only increased the wetted area by 12 grid cells on our

5  1166 x 2000 computational grid, whereas with 7 m SLR 3320 new ocean grid cells are generated. A detailed analysis of the impact of SLR on each tidal constituent is given below.



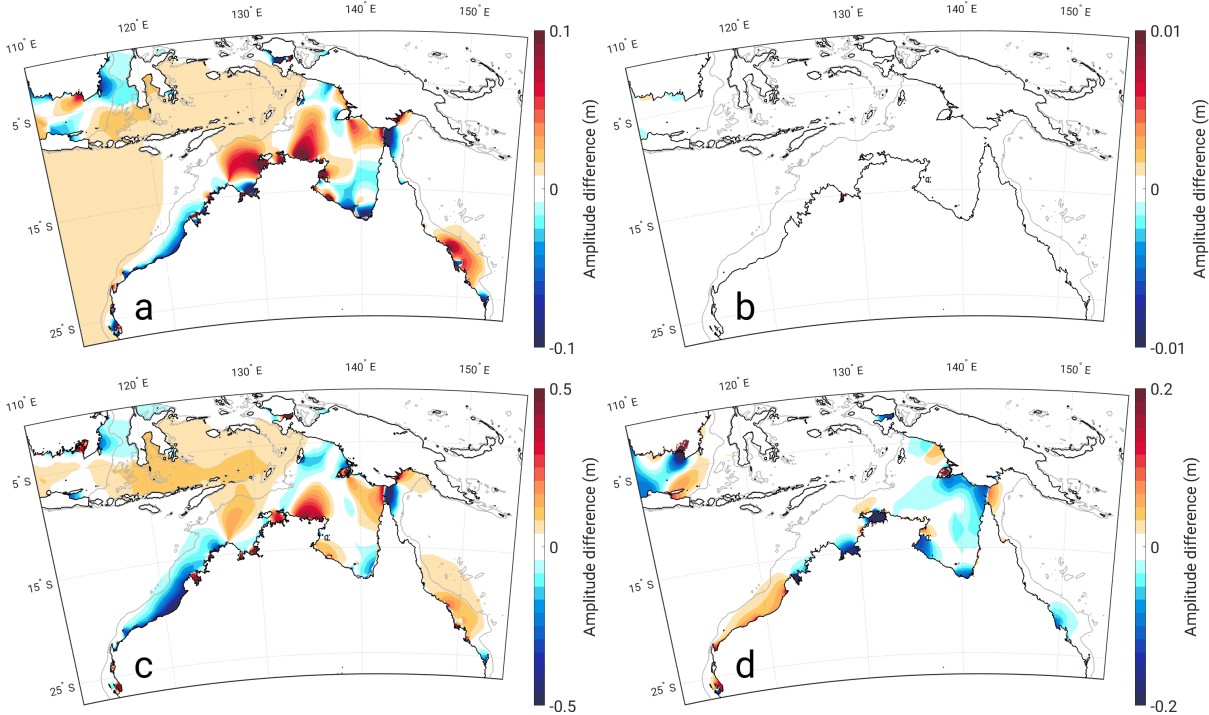

**Figure 4.** The difference in $M_2$ amplitude (m) between the FL and control simulation for 1 m SLR (a) and 7 m SLR (c) alongside the difference between the FL and NFL simulation for 1 m SLR (b) and 7 m SLR (d). Regions that appear red are where the FL amplitude is greater than that to which it is compared; as such coastal areas which have been flooded in b) and d) appear red.

## 3.1 Effect of SLR on the $M_2$ tidal constituent

Along Eighty-Mile beach $M_2$ undergoes a reduction with SLR, which is more pronounced in the NFL simulations than in the FL run; cf. Figs. 4c–4d. As a local feature in that area, the $M_2$ amplitude within King Sound ($\sim 1.2$ m) is notably smaller compared to the amplitude along the nearby coastline ($\sim 3$ m). This amplitude increases with SLR, but with retreating shorelines there is

5   substantial flooding at the head of the sound and the amount by which the amplitude increases is moderated.

Within Joseph Bonaparte Gulf (JBG), at small levels of SLR, there is a reduction in amplitude, contrasted with the Timor Sea to the north in which the $M_2$ response is slightly positive. These features are largely proportional in magnitude with SLR, whereas the channels in the south of JBG experience an amplification of tide. In the NFL simulation this increase in amplitude in the south of JBG extends northwards above 10 m SLR, but in the FL run there is flooding up Cambridge Gulf and Queens

10   channel, which dampens the tide and inhibits amplitude increases to the north. Opposite patterns are found in the Van Dieman Gulf (VDG), where $M_2$ amplitudes initially increase with low levels of SLR. However, when SLR exceeds 5 m, extensive flooding occurs on the eastern and southern shores of the gulf, with amplitude changes gradually switching their sign. By 10 m SLR, the amplitude has almost returned to present levels.



**Figure 5.** The constituent phases (deg) for the control (left) and 7 m SLR (right) simulations for $M_2$ (top), $S_2$ (middle), and $K_1$ (bottom)

The Arafura Sea experiences enhanced $M_2$ amplitudes with SLR, decreasing in magnitude away from the coast of Australia. The muted change in amplitude in the north of the Sea can be attributed to the movement of an amphidromic point (see Fig. 5) south off western New Guinea to become real by 7 m SLR, compare Fig. 5a with Fig. 5b. Note that this amphidrome forms earlier in the NFL simulations, at 5 m SLR. With varying shorelines, flooding occurs along the northern coast of Australia, attenuating the amplitude increase. The inundation of Yos Sudarso Island with higher levels of SLR comes with decreasing intensity of the amplitude response. Above 3 m SLR the phase lines directed north-south within the Gulf of Carpentaria coalesce to form two new amphidromic points (Fig. 5b). Correspondingly, amplitudes in the center of the Gulf are suppressed and the perturbations associated with SLR remain less than 10 cm.





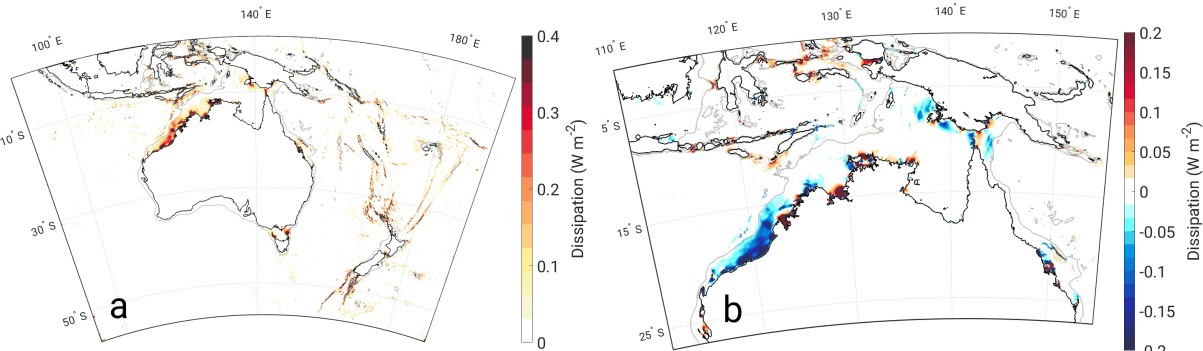

**Figure 6.** The dissipation (W m$^2$) associated with the M$_2$ constituent across the entire model domain (a) and the difference in dissipation between the 7 m FL simulation and the control run (b)

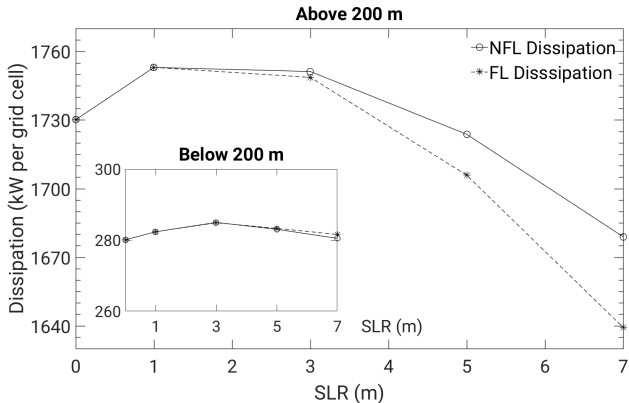

**Figure 7.** The dissipation totals across the model domain above the 200 m isobath and below it (inset figure). Note the difference in scale between the two panels.

Torres Strait features a sharp divide between positive and negative amplitude differences on west and east side respectively. The Strait is shallow and dotted with islands, restricting the flow of the tides. Amplitudes on the east side are elevated compared to the west but this difference is mitigated with SLR, as the larger volume of the channel allows for enhanced tidal transports across the strait.

5      Not shown in the figures are the effects of SLR on the tide around the south coast of Australia. These effects are generally smaller in magnitude than in the north (Schindelegger et al., 2018); with the notable exceptions of the Bass Strait, where the M$_2$ amplitude north of Tasmania is depressed with SLR, and both Gulf St. Vincent and Spencer Gulf, where the amplitude is increased with SLR.

     Figure 6a displays the dissipation associated with the M$_2$ tidal constituent for the control simulation. It is evident that a
10   majority of the energy loss occurs at the coast and on the shelf, especially around sharp and shallow bathymetric features. There is a striking reduction in dissipation along the north-west coast of Australia with SLR, matching areas with a marked



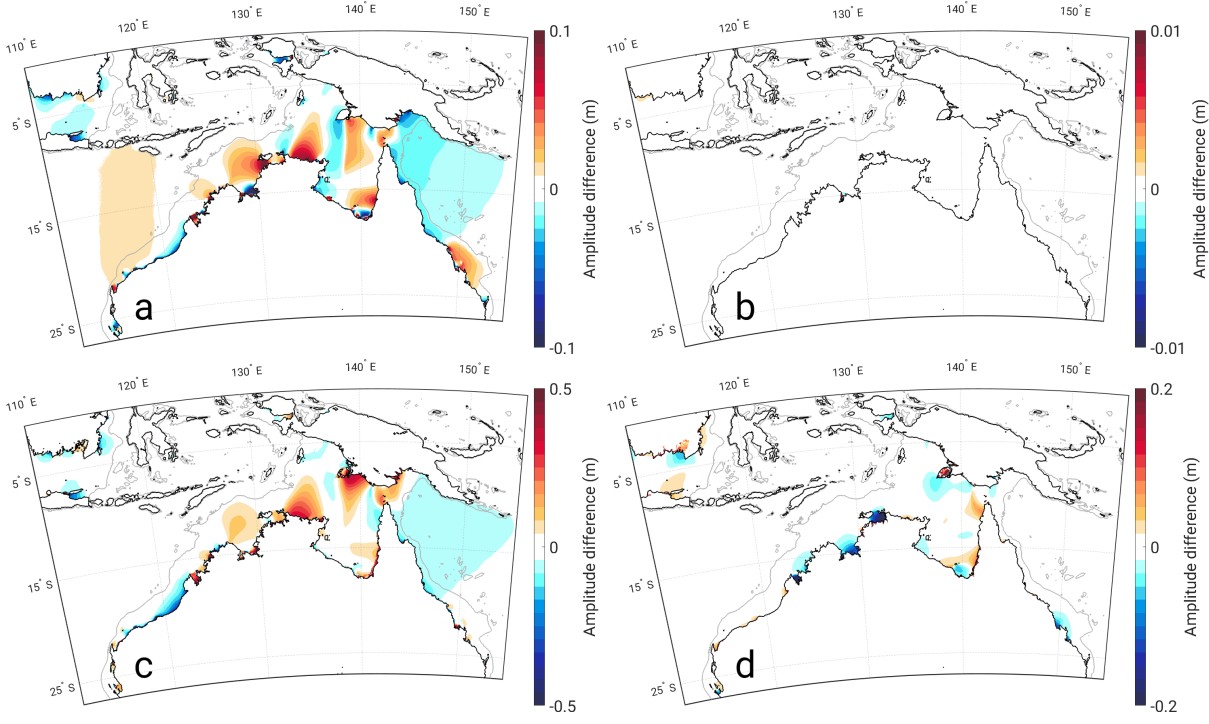

**Figure 8.** Same as Fig. 4 but for the $S_2$ constituent

decrease in tidal amplitude (Fig. 6b). Parts of King Sound and the south of JBG experience large increases in dissipation, as does VDG at the northern entrance in the NFL simulations; this is consistent with the increase in amplitude seen here in Fig. 4. This result is not shared by the runs with varying shorelines, as tidal energy is dissipated in newly flooded cells around the coast of the Gulf. Enhanced energy loss occurs through the channels in the island chain west of Timor as SLR increases to 5 m, after

5   which the dissipation through these channels begins to drop. The south coast of Papua, especially around the submerged Yos Sudarso island, concentrates dissipation with the appearance of widespread flooding in the area. Not shown in the figures are decreases in dissipation with SLR at the eastern and western entrances of the Bass strait. Overall, the dissipation per unit area on the shelf (nominally above 200 m in depth) far outweighs dissipation per unit area in the deep ocean, and peaks between 3 and 5 m SLR, before beginning to fall at higher sea levels (Fig. 7).

10  **3.2    Effect of SLR on the $S_2$ tidal constituent**

Figure 8 reveals 5–10 cm $S_2$ amplitude changes with 1 m SLR in certain areas along the north and north-west coast of Australia and the Arafura Sea. There are many similar features compared to how the $M_2$ amplitude changes with SLR, including a depression in amplitude along Eighty-mile Beach and an increase in amplitude in King Sound. In the NFL runs the amplitude in VDG increases greatly with SLR, but in the FL runs the magnitude of this response is reduced. The absence of significant





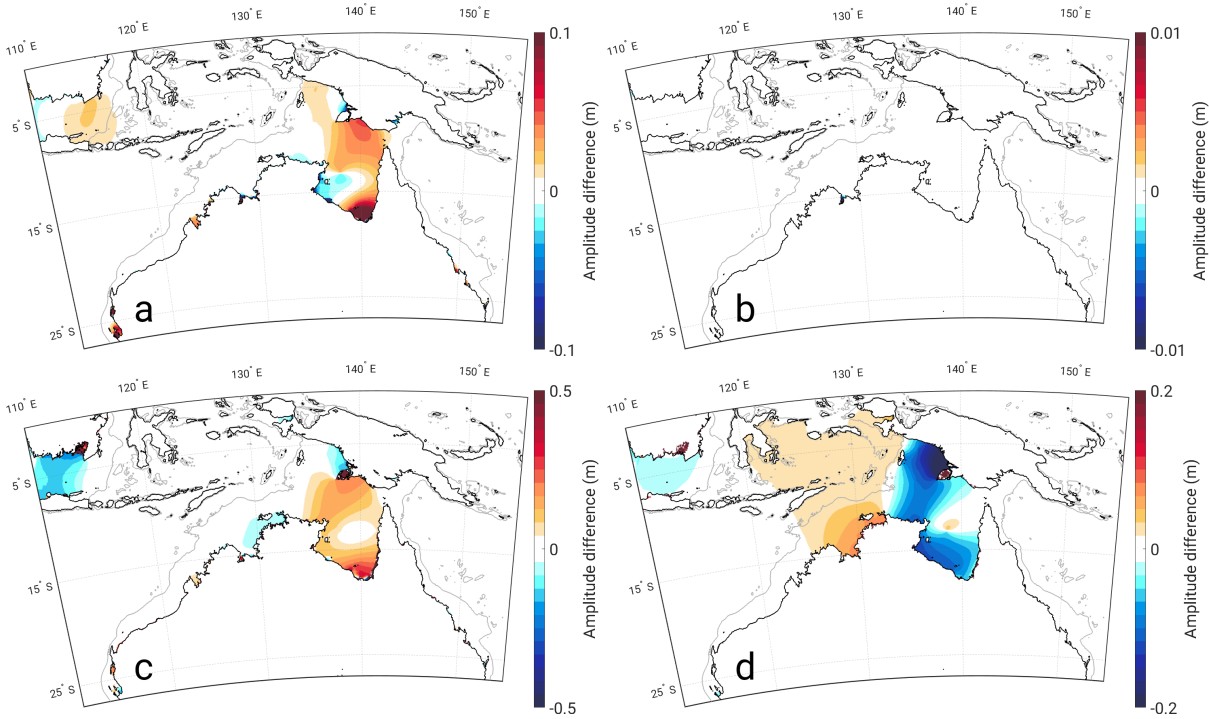

**Figure 9.** Same as Fig. 4 but for the $K_1$ constituent

signal across the Arafura Sea and Gulf of Carpentaria in Fig. 8d indicates the $S_2$ response in that area exhibits little sensitivity to the way shoreline positions are treated.

Figure 5 (panel c and d) highlights some of the changes to the $S_2$ constituent's phase with greater water depth. With $S_2$ being a semi-diurnal constituent, its response to SLR largely resembles that of $M_2$. For example, above 5 m SLR in the NFL scenario a new amphidromic point forms east of the Arafura Islands. However, in the FL simulations the phase lines that would converge to form this amphidrome diverge with increasing SLR. Unlike $M_2$, above 7 m SLR a degenerate amphidromic point on Tasmania moves offshore east towards New Zealand. Within the Gulf of Carpentaria there are two major amphidromes, one in the middle of the gulf, and one centered around Wellesley Island. In the NFL runs these points gradually move northwards with SLR, but in the inundation scenario they migrate southwards towards areas of flooding. Finally, we note (but do not explicitly show) that dissipation associated with the $S_2$ constituent has a striking spatial similarity to that of $M_2$, yet the $M_2$ contribution towards the total dissipation is four times that of $S_2$.

### 3.3 Effect of SLR on the $K_1$ tidal constituent

$K_1$ is fairly docile across the domain compared to the semi-diurnal constituents examined here. The main point of interest is the amphidromic system in the Gulf of Carpentaria. Figure 9 shows a large initial positive amplitude response in the south of the Gulf, while the amplitude in the western part decreases slightly. Above 1 m SLR this feature is consumed by an amplitude





increase completely surrounding the amphidrome in the center of the Gulf (Fig. 5e). The amphidrome moves slightly eastwards with SLR (Fig. 5f), presumably because of low-lying land near Carpentaria and the Wellesley Islands being inundated. In the FL simulations the region north of Yos Sudarso Island drops in amplitude, whereas in the NFL runs it shows a substantial positive response along with the rest of the Gulf of Carpentaria (Fig. 9c). The intensity of the positive and negative amplitude changes is also reduced in the FL simulations.

The majority of dissipation for which $K_1$ is responsible in the model domain (approximately seven times less than that of $M_2$) occurs East of the Arafura Islands and throughout the islands west of West Papua. From maps similar to Fig. 6 we have discerned that there is very little initial change in dissipation of $K_1$. Yet as higher SLR is assumed, dissipation drops in the south of the Gulf of Carpentaria, within JBG and west of the Arafura Islands. In the NFL simulations dissipation increases throughout the shelf and around the Gove peninsula. For runs with retreating shorelines dissipation drops around the islands west of West Papua, and there is a marked increase in energy losses over flooded areas in south Kalimantan and south Papua.

## 4 Discussion

It has been shown here, using a validated tidal model, that the tidal characteristics around Australia are sensitive to water column depth changes due to SLR. SLR has a broadly linear effect on the amplitude of the semi-diurnal constituents out on the open shelf, but causes increasingly large semi-diurnal amplitudes, and correspondingly high tidal dissipations, within embayments such as King Sound. As stated earlier, there was little difference in result between uniform SLR and SLR based on present-day sea-level trend data. When examining the effect of SLR of the magnitude we have shown it would be more fitting to use a "fingerprint" of sea level change, which includes elastic deformation, as shown in Wilmes et al. (2017, their Fig. 2). In this case, however, Australia sits away from any largely varying spatial signal, justifying our use of a uniform SLR.

One of the most striking characteristics of the response of the semi-diurnal constituents to SLR is the drop in amplitude along the north-west coast of Australia, an area where a large amount of dissipation occurs in the control. It is likely that altered propagation properties of the incoming tidal wave and an increase in dissipation around the islands west of Timor reduces the energy carried further onto the shelf by the tide. The wave further loses energy around VDG and JBG, and thus the amplitude along the north-west coast is reduced. The collapse of a semi-diurnal resonant response would explain the drop in total dissipation with large levels of SLR, as the semi-diurnal constituents are responsible for the majority of dissipation within the model. This is supported by the re-arrangement of the $M_2$ co-tidal lines between the control and higher SLR simulations (Fig. 5) as the system goes from hosting a single wave propagating across the Arafura Sea to a complex system with multiple amphidromes within the basin.

The movements of the amphidromes may be understood by considering the question presented by Taylor (1920). A semi-enclosed basin forced by a Kelvin wave, which is introduced at the open boundary and is allowed to reflect off the end, develops an amphidromic system. With the introduction of increasing dissipation due to flooding the amphidromes move towards the edge of the basin along which the reflected Kelvin wave travels; see Pelling et al. (2013, Fig. 2) for an illustration. A perfect example of this reorganization is seen for $S_2$ within the Gulf of Carpentaria (Fig. 5d). Two amphidromic points that are present





at 7 m SLR move towards the south-west when comparing the NFL and FL runs. The flooding along the south edge of the gulf increases the tidal dissipation there, reducing the energy of the tidal wave as it moves clockwise around the gulf, thus drawing the amphidromes towards the edge.

It appears that $K_1$ has an amplified response to SLR in the Gulf of Carpentaria region. The free period of oscillation of
an enclosed basin with length 500 km and depth 50 m, i.e., the approximate dimensions of the Gulf of Carpentaria, is about 27 hours. Increasing the water depth increases the propagation speed of the tide, and therefore reduces the free period of oscillation, bringing it closer to the diurnal period of the $K_1$ constituent and making resonance more likely.

This study has also shown that allowing the implementation of flooding can introduce areas of increased dissipation that can dampen or even cause the complete reduction of changes in amplitude due to SLR. The higher levels of SLR considered here,
though extreme, comprehensively show that allowing coastlines to flood with implemented SLR can have a significant impact to the amplitude and phase of the tides in the local area.

While the very high sea-level changes in this study may be useful for showing the impact of a large extent of flooding on tidal amplitudes, there are also model concerns as to how well the boundary conditions hold for the more extreme levels of SLR. Most of the boundary is out in the open ocean where tidal amplitudes are small and expansions of the water column
are virtually zero in relative terms. In such cases, any changes to changes in tidal amplitude with SLR will also be small. This argument may not be fully applicable to the domain boundary in the north, where it was computationally necessary to cut across the Sunda shelf. For future examinations of the Australasian domain, it is recommended either to extend the boundary north beyond Indonesia or to use a high resolution global tidal model to obtain boundary conditions from or to compare established boundary conditions to (e.g. Carless et al., 2016; Schindelegger et al., 2018). It is also recommended that a higher resolution
be used for the domain itself to more accurately resolve the small islands and coastal features.

With regards to other future work, a series of numerical tests, using forcing of different periods, is proposed to distinguish if the tidal response of certain areas within the domain is largely due to resonance or frictional effects (Idier et al., 2017). Additionally, an investigation into the coupling of tidal changes for expected magnitudes of SLR and storm surges (Muis et al., 2016) within the domain could be indispensable for providing insight and guidance to the future planning of coastal defenses
around Australia.

In conclusion, the series of simulations presented here have shown that the tidal amplitudes along the northern coast of Australia and around the Sahul shelf region are particularly sensitive to SLR. Coastal population centers such as Darwin, Mackay and Carpentaria are predicted to have to deal with the consequences of increased tidal amplitudes with increasing SLR. SLR appears to be moving the semi-diurnal constituents away from resonance on the shelf and the diurnal constituents
towards resonance in the Gulf of Carpentaria. The implementation of flooding can have a significant impact on the response of the tide by locally increasing dissipation, and should be considered essential for future tidal modeling with SLR.

*Acknowledgements.* Financial support for this study was made available by the UK Natural Environmental Research Council (grants NE/F014821/1 and NE/I030224/1, awarded to J.A.M.G) and the Austrian Science Fund (FWF, through project P30097-N29 awarded to





M.S.). The sea level data came from UHSLC (https://uhslc.soest.hawaii.edu/data/) and GESLA-2 (www.gesla.org). All numerical simulations were performed using HPC Wales' supercomputer facilities. Model results are available from the authors upon request.





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
