# Peer review of "The impact of sea-level rise on tidal characteristics around Australia"

_Ocean Science, 2018_

## Referee Comment (RC1) · Anonymous Referee #1 · 15 Oct 2018

SUITABILITY

The subject of the paper, i.e. the study of the impact of sea-level rise on tidal characteristics around Australasia falls within the fields covered by Ocean Sciences.

SUMMARY

The manuscript setup an established tidal model on the Australia area, focussing on the SLR effect on the Australasia area. A validation of the model is presented for present-day conditions. The SLR is implemented through a water depth increase. The study focusses on M2, S2 and K1 components changes, allowing the land to be flooded or not, while the investigated SLR values range from 1 to 20 m, and the paper focuses on the 1 and 7 m results. The model results for SLR=1 m are compared with tide gauge

processed data. Then, the effect of SLR on the tidal components around Australasia is described. Some elements of physical mechanisms are provided.

GENERAL COMMENTS

The present work, is, to my knowledge the first regional study of SLR effect on tides in this area (Australasia), even if global studies already provided indications (e.g. Pickering et al, 2017; Schindelegger et al., 2018). In addition, a well-established model is used, together with published methods of analysis. This make this paper suitable for a potential publication. However, the paper lacks a description of the present-day tidal dynamics. The model validation lacks some elements to be fully convincing. In addition, the comparison of the observed trend and modelled trend rises questions on the SLR value choice. The paper is sometime difficult to read (especially when describing results per areas not indicated in maps). The provided physical explanations of the results deserve to be more strongly supported. Some figures and maps are probably lacking (regarding the text), and deserve to be in an Appendix.

MAJOR REMARKS

1.Site description

The paper lacks description on the tidal dynamics in the study site, making the results more difficult to interpret. A minimum level of description should be provided. Maps of M2, S2, K1 amplitudes (and perhaps phase) would be useful.

2.Model validation

Regarding the validation of the amplitude of M2, S2, K1, statistical information as correlation coefficient and bias would allow to better characterize the model errors, and also to better support the text. In particular, it is stated that the model overestimates K1. An explanation is given. But, looking at Figure 2, I have the impression that S2 is also over-estimated. If this is the case, then the explanation would not be reliable anymore. In addition, there is no physical explanations provided for the sites which are

beyond the 2+/- standard deviation and these sites are not identified (we do not know where the model "fails"). Regarding the comparison between the modeled trend and observed trend, I have a concern on the SLR choice. Indeed, I do not understand why using SLR=1m, rather than a more probable value for the last decades. The underlying assumptions (not stated in section 2.4, but stated later on) is that the changes are proportional to SLR. While this has been proved to be true in some locations, this is can be locally not true. The validation of trends deserve more attention, either by checking the proportionality of changes in the [0-1m] or using a more realistic SLR value (or a non-uniform SLR field) for the last decades. As to me Figure 3 and the text is not fully convincing, I strongly recommend to have a closer look on this point. In addition, the validation should also be done for S2 and K1.

3.Physical mechanisms

Several times in the paper, the authors provide some explanation on the results (quality or SLR effect) using the words "probably", "presumably". This weakens the paper. As much as possible, the authors should provide more evidence to support their interpretations. As written in the discussion a series of numerical tests could be done to better asses the resonance and frictional effects. I strongly recommend to perform these experiments in the present paper to really support the interpretations. As a more minor remark, the model does not include advection terms. What could be the effect of neglecting this term on the present results? Is there any literature justifying to neglect it for tide modeling?

4.Figures

- Maps of M2, S2, K1 amplitudes are lacking.

- The text relies on many results, which are not shown (e.g. tide changes of M2, S2, K1 for SLR different that the 1 and 7 m shown in the paper). Such figures would be useful and could be added in Appendix.

- The text describes the results using the names of many locations. A map indicating all this locations is needed (a reader not knowing Australia will have to make a big effort to follow the description).

- In the text, there are also some comments on tide changes south of Australia. Some figures to support this text would be useful, in appendix for instance.

"ON-LINE" REMARKS

- P1-Line 14: sentence "At sea level . . .." is a bit strange. Why insisting on well-suited farming?

- P1-Line 16: provide a number together with the 85% would be more meaningful

- P2-Line 12: Pickering et al., 2012 -> Pickering et al., 2017

- P5-Line 7: why focusing on M2, S2, K1? Some explanations should be provided. Perhaps they are the dominant tidal components but it should be said (relying on reference or map?).

- P5-Line 25: as the authors made the computation under non-uniform SLR, this would be useful/interesting to add in appendix the tide changes induced considering the non-uniform SLR.

- P7-Line 3-5: "These statistics . . .". The authors do not provide enough evidence that this is the spatial resolution that could explain the discrepancies. More detailed analysis is required to support this hypothesis.

- P8-Line 2-3: remind that this was for a given range of SLR in "Idier et al. (2017)", and also for a given area (NW European shelf).

- P14-Line 14: "SLR has a broadly linear effect on the amplitude of the semi-diurnal constituents out on the open shelf, but causes increasingly large semi-diurnal amplitudes, and correspondingly high tidal dissipations, within embayments such as King Sound". I did not see "the linear" effect on the figures. Looking at Figure 4, 8 and 9, notable differences can be observed offshore between the two SLR scenarios. This point deserves more explanation, and probably some kind of maps showing proportionality coefficients of tide changes with SLR, as for instance in (Pickering et al., 2017) or in (Idier et al., 2017).

- P15-Line 16: why was it computationally necessary to cross the shelf? Are the authors referring here to computational time? If yes, then it should be stated more clearly and computation time should be provided. In addition, one simulation on a larger domain for a very large SLR would allow estimating the effect of the assumption that tidal components are unchanged on this north boundary.

---

## Referee Comment (RC2) · Anonymous Referee #2 · 17 Oct 2018

I recommend major revisions for this manuscript. The topic, tidal changes around the Australasia region, is interesting. The study adds to the literature on projected tidal changes in response to sea level rise with a high-resolution model of the Australasia region. As far as I can tell, the study has been done competently. The writing is generally clear, with some exceptions noted below.

The main cause for my concern is that, as far as I can tell, the authors have used TPXO boundary conditions throughout their study. The TPXO boundary conditions are from the present day, meaning that the tides along the boundaries are not responding to changes in sea level rise. The fact that the regional model has some skill in simulating observed tidal changes suggests that maybe this is OK. On the other hand, the authors have a high-resolution forward global tide model available to them; why didn't they use

[Figure]

it here? Global tide models would respond to the changes in sea level, thus providing more natural boundary conditions. Would the computational expense be too great? If so, say so, and provide some evidence for that, or at least make it more clear that we shouldn't worry too much about this. If it is feasible, I suggest that the authors use the global tide models to complement at least some of the simulations with TPXO boundary conditions.

Other important suggestions:

1) Where does the SAL term come from in this regional model? This is an important detail, that should be described.

2) Page 5, lines 20-29: The 1 and 7 meter sea level rise values are justified, but the 3, 5, 15, and 20 meter sea level rise values are not explicitly justified. 3 and 5 lies between 1 and 7, the latter being an "extreme value" so I'm guessing that might justify the 3 and 5 meter values; but again, it would be nicer if the authors themselves made an explicit justification. And the 15 and 20 meter values are not justified at all.

3) It seems to me that readers would take more away from the discussion of Figure 1 if the tide trends were compared to the MSL trends/increases. Are the tidal trends comparable? Other papers e.g. Jay 2009 have commented on this-in some regions, the tidal and MSL trends are comparable. This helps the readers to envision the societal significance of the tidal trends. I suggest adding some commentary on this for the Australasia region.

Minor comments:

Page 6 line 4-suggest "With a few exceptions, record lengths are short, but all . . ."

Page 6 line 20-suggest ". . .constituent amplitudes. . ." in place of ". . .constituents amplitudes. . ."

Page 7, line 29-can the stated greater impact of sea level rise on tides be justified with a citation or some other source of information?

Page 9, line 8-suggest "amplification of the tide" (insert "the")

Figure 6 caption – "W mˆ2" i.e. should instead be "W mˆ-2"

Page 12, line 12-suggest "…Arafura Sea. The changes in S2 amplitude appear similar to the changes in M2 amplitude, including…"

Page 13, lines 3-5-suggest "Because S2 is a tidal constituent, its response…"

Paige 13, line 6-suggest "In contrast to the M2 behavior, above 7 m…"

Page 13, line 10-the phrase beginning with "yet" sounds odd to me. The M2 and S2 dissipation patterns are similar but the M2 values are much larger. That is not surprising, so inserting a phrase beginning with "yet" seems out-of-place, to me at least. Minor point, but I suggest omitting this phrase.

Page 13, line 13-suggest "The K1 changes are relatively limited compared to the changes in the semi-diurnal constituents examined here."

Page 14, line 6-this sentence reads awkwardly. Please improve the grammar.

Page 14, line 17-I believe that the word after "SLR" should be "on" not "of"

Page 15, lines 10-11-"impact to" should be "impact on"

Page 15, line 13-"model concerns" is an odd-sounding phrase

Page 15, line 15-"changes to changes". Is this what you want to say?

---

## Editor Comment (EC1) · P.L. Woodworth (Editor) · 24 Oct 2018

Additional comments on 'The impact of sea-level rise on tidal characteristics around Australasia' by Harker et al. (OSD)

I read this paper with interest and have a few comments additional to those of the two formal reviewers.

A first is the word Australasia in the title. Australasia means Australia, New Zealand and the west Pacific islands and (maybe) Papua New Guinea. But there is no discussion of New Zealand tides in the text, so at first I thought Australasia should be replaced by Australia. But then many of the figures even cut off the southern part of Australia. Why was that? So I think the title might be revisited.

[Figure]

page 1, line 5 and 8 - there are mentions in the abstract and text of places that can be unfamiliar and so need qualifying e.g. on line 5 this should be Arafura Sea (between Australia and Papua New Guinea). On line 8 Papua should be Papua New Guinea I guess (Papua is a province of Indonesia which I think is not is what is meant). Some of these places are later pointed out in Figure 1 but the reader will not know them at this point.

22 - I don't understand why Woodworth (2017) is given here. It is not relevant to the sentence.

page 2, line 1 - the main peaks in extremes in most parts of the ocean, where there is a semidiurnal tide, are every 4.4 years or so from the perigean cycles in the moon's orbit. You get peaks in extremes every 18.6 years where there are diurnal tides. You could references Haigh (JGR, 2011) for example or Pugh and Woodworth (2014) or Merrifield et al. (JGR, 2013).

10 - again I don't see why the Mawdsley reference is relevant to this sentence.

13-15 - you could reference the AR5 somewhere here.

23 - phenomena –> phenomenon

24 - again, who knows where the Sahul shelf is?

26 - dissipation on a par with the Yellow Sea ..

page 5, eqs. 6 and 7 - these need reformatting

23 ' 'Additional runs'. I think a few extra words are needed to clarify that these additional runs were not used.

page 6, Figure 2 - ok for amplitude. Is phase lag agreement worth showing?

para at line 11 - is it worth mentioning how long-period tides come into this?

20 - reword 'A comparison of the amplitudes of the constituents calculated ...'

page 7, line 18 - I can see from the figure that the signs are often in agreement, I can't see the 'reproduces much of the in situ variability'. Needs explaining better.

page 8, line 3 - surely standard deviations should be standard errors?

.... annual tidal estimates of M2. Stations with insignificant measured phase ...

section 3 - this seems to me to need a couple of introduction sentences to say that you will here in this section be testing SLR of 1,3,7 m for the modelling.

page 9, 13 - I don't see how the reader can relate to 10m change which is not shown, so add (not shown) to make it clear.

page 10, in Fig 5 caption and the y-annotation 'phase' should be 'phase lag' and it is Greenwich phase lag presumably.

line 2 - I think this should read:

.. movement of a virtual amphidromic point (an amphidrome over land) (see Fig. 5) .. to become real (i.e. amphidrome over the ocean) ...

page 11, line 1 of Figure 6 caption - .. model domain in the control run.

You should make clear which way the difference works.

Figure 7 - does the 200m refer to the control run bathymetry?

second sentence caption - I don't think is necessary.

line 5 - why south coast chopped off?

page 12, 1 and 2 - have JBG and VDG acronyms been defined?

13 - Eighty-Mile Beach

page 13, Figure 9 - you have FL-cntl and FL-NFL so it might help to have a third column of maps for NFL-cntl so the reader did not have to do mental arithmetic.

[Figure]

line 13 - docile is an odd word. Just say K1 has a small amplitude (< ?? cm)

page 14, 4 - Figure 9c must be 9d

23 - reduce

page 15, 11 - in the amplitude

14-16 - I don't understand this sentence but anyway 'changes to changes' needs re-wording.
* * *

---

## Author Comment (AC1) · 15 Jan 2019

We thank the reviewer for their constructive commentary about our work. Comments received about the implementation of the boundary conditions of the domain prompted a review and recalculation of our simulations, resulting in significantly different results. We moved from using elevations as prescribed by TPXO8 to those from a global forward model. For computational cost and time reasons this model was run using the M2 and K1 constituents only, making it necessary to limit the manuscript results to these constituents. Unfortunately this means our discussion of S2 no longer appears in the manuscript, and any suggestions pertaining to this section cannot be addressed.

*GENERAL COMMENTS The present work, is, to my knowledge the first regional study*

[Figure]

*of SLR effect on tides in this area (Australasia), even if global studies already provided indications (e.g. Pickering et al, 2017; Schindelegger et al., 2018). In addition, a well-established model is used, together with published methods of analysis. This make this paper suitable for a potential publication. However, the paper lacks a description of the present-day tidal dynamics. The model validation lacks some elements to be fully convincing. In addition, the comparison of the observed trend and modelled trend rises questions on the SLR value choice. The paper is sometime difficult to read (especially when describing results per areas not indicated in maps). The provided physical explanations of the results deserve to be more strongly supported. Some figures and maps are probably lacking (regarding the text), and deserve to be in an Appendix.*

*MAJOR REMARKS 1.Site description The paper lacks description on the tidal dynamics in the study site, making the results more difficult to interpret. A minimum level of description should be provided. Maps of M2, S2, K1 amplitudes (and perhaps phase) would be useful.*

The figure showing the bathymetry of the domain has been expanded to include a chart of M2, and a supplemental figure of K1 has also been included. We hope the additional references to this figure in the text and the expanded description of the control simulation in the Results section [page 10, line 4] help the reader to visualise the changes we describe.

*2.Model validation Regarding the validation of the amplitude of M2, S2, K1, statistical information as correlation coefficient and bias would allow to better characterize the model errors, and also to better support the text. In particular, it is stated that the model overestimates K1. An explanation is given. But, looking at Figure 2, I have the impression that S2 is also over-estimated. If this is the case, then the explanation would not be reliable anymore. In addition, there is no physical explanations provided for the sites which are beyond the 2+/- standard deviation and these sites are not identified (we do not know where the model "fails"). Regarding the comparison between the modeled trend and observed trend, I have a concern on the SLR choice. Indeed, I*

*do not understand why using SLR=1m, rather than a more probable value for the last decades. The underlying assumptions (not stated in section 2.4, but stated later on) is that the changes are proportional to SLR. While this has been proved to be true in some locations, this is can be locally not true. The validation of trends deserve more attention, either by checking the proportionality of changes in the [0-1m] or using a more realistic SLR value (or a non-uniform SLR field) for the last decades. As to me Figure 3 and the text is not fully convincing, I strongly recommend to have a closer look on this point. In addition, the validation should also be done for S2 and K1.*

This comment contains several threads. Our responses to each of those threads are as follows:

(1) Validation of the M2 and K1 control run and discussion of outliers: we have reworked Section 2.4 [page 8, line 5-10] to explain model-to-data discrepancies beyond the 1-sigma-level (stations Williamstown and Geelong). Correlation coefficients and median absolute differences have been included as additional statistical measures. Note that upon exclusion of the two "outliers", RMS differences improve significantly w.r.t. the previous version of the manuscript.

(2) Possible overestimation of K1: even though our new Figure 2 contains slight hints of such an overestimation, we have refrained from speculations in this regard.

(3) SLR choice: we tried to redo both our new global and the Australian domain simulations with a few dm of SLR, but obtained suspicious tidal changes, possibly related to numerical artifacts arising in the model at such small values of SLR. Therefore, the validation is still done with the M2/K1 results for 1 m.

(4) Validation of trends for constituents other than M2: we have performed the necessary computations for K1 and show the corresponding figure in the supplement. Finally, we note that alongside SLR, other processes (e.g., ocean warming, thermocline deepening, changes in shoreline position, local anthropogenic influences such as periodic dredging) could have contributed to observed changes in the tides, so a comparison to

model values from SLR-only scenarios can't be fully convincing. Figure 3 rather documents partial success in capturing observed tidal variability at a number of stations. Additional formulations pertaining to this issue have been added to the text in Section 2.4.

*3.Physical mechanisms Several times in the paper, the authors provide some explanation on the results (quality or SLR effect) using the words "probably", "presumably". This weakens the paper. As much as possible, the authors should provide more evidence to support their interpretations. As written in the discussion a series of numerical tests could be done to better asses the resonance and frictional effects. I strongly recommend to perform these experiments in the present paper to really support the interpretations. As a more minor remark, the model does not include advection terms. What could be the effect of neglecting this term on the present results? Is there any literature justifying to neglect it for tide modeling?*

We have followed the first suggestion and removed many of the conjectures in the original manuscript. With the efforts and textual changes demanded by this revision and the new simulations, we think that more numerical tests fall beyond the scope of our study. As far as the advection terms are concerned, these can be neglected with little drop in accuracy (as per Egbert et al., 2004), with the added benefit of reducing the computational workload of the model runs. Also, test runs by one of the authors (M. Schindelegger) with another tidal model showed that M2 responses around Australia to a 2 m uniform SLR change by no more than 2 mm between simulations with and without advective terms.

*4.Figures - Maps of M2, S2, K1 amplitudes are lacking.*

Maps of M2 and K1 amplitudes are now included alongside the bathymetry and in the supplementary information respectively.

*- The text relies on many results, which are not shown (e.g. tide changes of M2, S2, K1 for SLR different that the 1 and 7 m shown in the paper). Such figures would be*

*useful and could be added in Appendix.*

The text is now more focused on what is shown in the figures and does no more rely on results for SLR other than 1 and 7 m. Note also that there are only slight variations in the spatial distribution of the tide changes for SLR different to 1 m and 7 m; we have therefore refrained from including these results in the appendix (or supplement).

*- The text describes the results using the names of many locations. A map indicating all this locations is needed (a reader not knowing Australia will have to make a big effort to follow the description).*

Figure 1 now includes markers showing the location of discussed topographical features, and labels naming specific bodies of water.

*- In the text, there are also some comments on tide changes south of Australia. Some figures to support this text would be useful, in appendix for instance.*

Figure 5, showing the south coast of Australia, has been included in the manuscript.

*"ON-LINE" REMARKS - P1-Line 14: sentence "At sea level . . .." is a bit strange. Why insisting on well-suited farming?*

The wording has been altered to a more general description of the suitability of coasts for human settlement, rather than a specific example. Reference to "At sea level" has been changed to "Coastal areas"

*- P1-Line 16: provide a number together with the 85% would be more meaningful*

A population number has been included alongside the population percentage: "85% of the population of Australia (approximately 19.9 million people; ABS, 2016)"

*P2-Line 12: Pickering et al., 2012 -> Pickering et al., 2017*

Modified as suggested by the reviewer

*- P5-Line 7: why focusing on M2, S2, K1? Some explanations should be provided.*

*Perhaps they are the dominant tidal components but it should be said (relying on reference or map?).*

Our new simulations focus on only M2 and K1 which are the dominant semi-diurnal and diurnal constituents in the domain which is now referenced to in Section 2.1. We have included Figures showing the control M2 and K1 amplitudes.

*- P5-Line 25: as the authors made the computation under non-uniform SLR, this would be useful/interesting to add in appendix the tide changes induced considering the non-uniform SLR.*

A figure showing the difference in result between uniform and non-uniform SLR has been included in the supplement, discussing our investigation into using spatially non-uniform SLR patterns

*- P7-Line 3-5: "These statistics . . .". The authors do not provide enough evidence that this is the spatial resolution that could explain the discrepancies. More detailed analysis is required to support this hypothesis.*

New analysis shows good agreement with all stations apart from Williamstown and Geelong, which are unique in that they are confined within Port Phillip Bay. This narrow headlands which confine this bay are too small a feature to be captured by the model resolution. How this area is treated in the model is now discussed.

*- P8-Line 2-3: remind that this was for a given range of SLR in "Idier et al. (2017)", and also for a given area (NW European shelf).*

Modified as suggested by the reviewer.

*- P14-Line 14: "SLR has a broadly linear effect on the amplitude of the semi-diurnal constituents out on the open shelf, but causes increasingly large semi-diurnal amplitudes, and correspondingly high tidal dissipations, within embayments such as King Sound". I did not see "the linear" effect on the figures. Looking at Figure 4, 8 and 9, notable differences can be observed offshore between the two SLR scenarios. This point*

*deserves more explanation, and probably some kind of maps showing proportionality coefficients of tide changes with SLR, as for instance in (Pickering et al., 2017) or in (Idier et al., 2017).*

With the new simulations we have performed this point of ours is no longer valid and has been removed from the text.

*- P15-Line 16: why was it computationally necessary to cross the shelf? Are the authors referring here to computational time? If yes, then it should be stated more clearly and computation time should be provided. In addition, one simulation on a larger domain for a very large SLR would allow estimating the effect of the assumption that tidal components are unchanged on this north boundary.*

This point is has been addressed by the new simulations and the formulations in question have been removed from the text.

---

## Author Comment (AC2) · 15 Jan 2019

We thank the editor for their constructive commentary about our work. Comments received about the implementation of the boundary conditions of the domain prompted a review and recalculation of our simulations, resulting in significantly different results. We moved from using elevations as prescribed by TPXO8 to those from a global forward model. For computational cost and time reasons this model was run using the M2 and K1 constituents only, making it necessary to limit the manuscript results to these constituents. Unfortunately this means our discussion of S2 no longer appears in the manuscript, and any suggestions pertaining to this section cannot be addressed.

*A first is the word Australasia in the title. Australasia means Australia, New Zealand and*

[Figure]

the west Pacific islands and (maybe) Papua New Guinea. But there is no discussion of New Zealand tides in the text, so at first I thought Australasia should be replaced by Australia. But then many of the figures even cut off the southern part of Australia. Why was that? So I think the title might be revisited.

The title has been revised in line with your comment. The previous focus on the north of Australia was because that was where most of the notable changes to amplitude within the domain occurred. As of the new simulations undertaken, a new figure showing the south has been included.

page 1, line 5 and 8 - there are mentions in the abstract and text of places that can be unfamiliar and so need qualifying e.g. on line 5 this should be Arafura Sea (between Australia and Papua New Guinea). On line 8 Papua should be Papua New Guinea I guess (Papua is a province of Indonesia which I think is not is what is meant). Some of these places are later pointed out in Figure 1 but the reader will not know them at this point.

The abstract has been rewritten with, hopefully, more immediately recognisable descriptions of locations.

22 - I don't understand why Woodworth (2017) is given here. It is not relevant to the sentence.

This was a mistaken placement of the reference, now removed

page 2, line 1 - the main peaks in extremes in most parts of the ocean, where there is a semidiurnal tide, are every 4.4 years or so from the perigean cycles in the moon's orbit. You get peaks in extremes every 18.6 years where there are diurnal tides. You could references Haigh (JGR, 2011) for example or Pugh and Woodworth (2014) or Merrifield et al. (JGR, 2013).

10 - again I don't see why the Mawdsley reference is relevant to this sentence.

13-15 - you could reference the AR5 somewhere here.

*23 - phenomena –> phenomenon*

The above spelling and reference suggestions have been implemented

*24 - again, who knows where the Sahul shelf is?*

Potentially confusing place names have been removed, and the locations discussed in the paper are now all found in Figure 1.

*26 - dissipation on a par with the Yellow Sea ..*

Changed to "...dissipation comparable to that of the Yellow Sea..."

*page 5, eqs. 6 and 7 - these need reformatting*

*23 ' 'Additional runs'. I think a few extra words are needed to clarify that these additional runs were not used.*

The Section 2.3 has been reformulated to describe the new simulations performed.

*page 6, Figure 2 - ok for amplitude. Is phase lag agreement worth showing?*

Median phase differences have been calculated and describe in the text [page 8, line 15]

*20 - reword 'A comparison of the amplitudes of the constituents calculated ...'*

Reworded to "A regression of the constituent amplitudes calculated"

*page 7, line 18 - I can see from the figure that the signs are often in agreement, I can't see the 'reproduces much of the in situ variability'. Needs explaining better.*

*page 8, line 3 - surely standard deviations should be standard errors?*

*.... annual tidal estimates of M2. Stations with insignificant measured phase . . .*

Section rewritten to account for above comments

*section 3 - this seems to me to need a couple of introduction sentences to say that you*

*will here in this section be testing SLR of 1,3,7 m for the modelling.*

*page 9, 13 - I don't see how the reader can relate to 10m change which is not shown, so add (not shown) to make it clear.*

Section has been rewritten to discuss the updated simulation.

*page 10, in Fig 5 caption and the y-annotation 'phase' should be 'phase lag' and it is Greenwich phase lag presumably.*

Caption and Figure altered

*line 2 - I think this should read: .. movement of a virtual amphidromic point (an amphidrome over land) (see Fig. 5) .. to become real (i.e. amphidrome over the ocean) . . .*

Modified as suggested by comment

*page 11, line 1 of Figure 6 caption - .. model domain in the control run. You should make clear which way the difference works.*

The way the difference works is now explicitly stated in the figure caption "(FL - Control)"

*Figure 7 - does the 200m refer to the control run bathymetry?*

Yes, this has now been made explicit in the figure caption.

*second sentence caption - I don't think is necessary.*

Second sentence in caption has been removed

*line 5 - why south coast chopped off?*

This was to highlight some of the more complex structure in the dissipation changes. The figure now shows the full domain.

*page 12, 1 and 2 - have JBG and VDG acronyms been defined?*

The acronyms have been deemed unnecessary and removed.

*13 - Eighty-Mile Beach*

Sentence has been removed

*page 13, Figure 9 - you have FL-cntl and FL-NFL so it might help to have a third column of maps for NFL-cntl so the reader did not have to do mental arithmetic.*

The figures originally showed FL-Ctrl beside NFL-Ctrl, however the differences were very difficult to tell by eye – hence why we show NFL-FL instead, which more easily presents the difference. The trade-off is the mental arithmetic, hopefully aiding by the text. I believe a third column would limit the ability of the figures to show some of the smaller details.

*line 13 - docile is an odd word. Just say K1 has a small amplitude (< ?? cm)*

*page 14, 4 - Figure 9c must be 9d*

*23 - reduce*

*page 15, 11 - in the amplitude*

*14-16 - I don't understand this sentence but anyway 'changes to changes' needs re-wording.*

Modified the above as suggested

---

## Author Comment (AC3) · 15 Jan 2019

*I recommend major revisions for this manuscript. The topic, tidal changes around the Australasia region, is interesting. The study adds to the literature on projected tidal changes in response to sea level rise with a high-resolution model of the Australasia region. As far as I can tell, the study has been done competently. The writing is generally clear, with some exceptions noted below. The main cause for my concern is that, as far as I can tell, the authors have used TPXO boundary conditions throughout their study. The TPXO boundary conditions are from the present day, meaning that the tides along the boundaries are not responding to changes in sea level rise. The fact that the regional model has some skill in simulating observed tidal changes suggests that maybe this is OK. On the other hand, the authors have a high-resolution forward global*

[Figure]

*tide model available to them; why didn't they use it here? Global tide models would respond to the changes in sea level, thus providing more natural boundary conditions. Would the computational expense be too great? If so, say so, and provide some evidence for that, or at least make it more clear that we shouldn't worry too much about this. If it is feasible, I suggest that the authors use the global tide models to complement at least some of the simulations with TPXO boundary conditions.*

We thank the reviewer for their constructive commentary about our work. Comments received about the implementation of the boundary conditions of the domain prompted a review and recalculation of our simulations, resulting in significantly different results. We moved from using elevations as prescribed by TPXO8 to those from a global forward model. For computational cost and time reasons this model was run using the M2 and K1 constituents only, making it necessary to limit the manuscript results to these constituents. Unfortunately this means our discussion of S2 no longer appears in the manuscript, and any suggestions pertaining to this section cannot be addressed.

*Other important suggestions:*

*1) Where does the SAL term come from in this regional model? This is an important detail, that should be described.*

The SAL term is derived from TPXO8 data. This has been clarified in Section 2.1.; "The model solves equations 1–2 using forcing from the astronomic tide generating potential only...and the SAL term is derived from TPXO8"

*2) Page 5, lines 20-29: The 1 and 7 meter sea level rise values are justified, but the 3, 5, 15, and 20 meter sea level rise values are not explicitly justified. 3 and 5 lies between 1 and 7, the latter being an "extreme value" so I'm guessing that might justify the 3 and 5 meter values; but again, it would be nicer if the authors themselves made an explicit justification. And the 15 and 20 meter values are not justified at all.*

Our new simulations have used 1, 3, 5, 7 and 12 m SLR. The 3 and 5 m values follow

from Wilmes et al., 2017, (as the global mean sea level increase from the collapse of just the marine sectors of the West Antarctic Ice Sheet and the total collapse of the WAIS respectively), which is now stated in Section 2.3. Whilst 12 m may not be physically justified, it allows us to see how trends may continue; this has been stated in the text.

*3) It seems to me that readers would take more away from the discussion of Figure 1 if the tide trends were compared to the MSL trends/increases. Are the tidal trends comparable? Other papers e.g. Jay 2009 have commented on this-in some regions, the tidal and MSL trends are comparable. This helps the readers to envision the societal significance of the tidal trends. I suggest adding some commentary on this for the Australasia region.*

This is a good point. A brief discussion of this has been added to the introduction.

*Minor comments:*

*Page 6 line 4-suggest "With a few exceptions, record lengths are short, but all . . ."*

*Page 6 line 20-suggest ". . .constituent amplitudes. . ." in place of ". . .constituents amplitudes. . ."*

The above modifications suggested by the reviewer have been made where possible (some sentences referred to no longer exist in the manuscript)

*Page 7, line 29-can the stated greater impact of sea level rise on tides be justified with a citation or some other source of information?*

This comment was written with a current review paper by Haigh et al. (submitted to Reviews of Geophysics) in mind. We could cite is as "gray literature" here but rather prefer to wait, see how this reference paper evolves, and include it later during the editing process.

*Page 9, line 8-suggest "amplification of the tide" (insert "the")*
*Figure 6 caption – "W mӞĘ2" i.e. should instead be "W mӞĘ-2"*

*Page 12, line 12-suggest ". . .Arafura Sea. The changes in S2 amplitude appear similar to the changes in M2 amplitude, including. . ."*

*Page 13, lines 3-5-suggest "Because S2 is a tidal constituent, its response. . ."*

*Paige 13, line 6-suggest "In contrast to the M2 behavior, above 7 m. . ."*

*Page 13, line 10-the phrase beginning with "yet" sounds odd to me. The M2 and S2 dissipation patterns are similar but the M2 values are much larger. That is not surprising, so inserting a phrase beginning with "yet" seems out-of-place, to me at least. Minor point, but I suggest omitting this phrase.*

*Page 13, line 13-suggest "The K1 changes are relatively limited compared to the changes in the semi-diurnal constituents examined here."*

*Page 14, line 6-this sentence reads awkwardly. Please improve the grammar.*

*Page 14, line 17-I believe that the word after "SLR" should be "on" not "of"*

*Page 15, lines 10-11-"impact to" should be "impact on"*

*Page 15, line 13-"model concerns" is an odd-sounding phrase*

*Page 15, line 15-"changes to changes". Is this what you want to say?*

The above modifications suggested by the reviewer have been made where possible (some sentences referred to no longer exist in the manuscript)

---

## Editor Comment (EC2) · Woodworth (Editor) · 16 Jan 2019

I believe the authors have responded well to the comments of the 2 reviewers and to my additional remarks, so I would encourage them to progress to a revised version as soon as possible. I am a bit concerned that the new work that the authors have done, and the scale of the changes implied in their responses, may mean that the revised version is substantially different to the OSD version, but that cannot be avoided.
* * *

---

## Author Response (AR2)

**Response to the Topic Editor**

I am happy that the authors have responded well to the comments of the reviewers and that this is now a better paper which will be read by many people interested in tides. I read through it again and have only two further minor remarks. One is that that 'astronomical tide' would be better than 'astronomic tide' which to some people means a gigantic tide. Also, to be pedantic, you refer in the text to Figure 6 before 4 and 5 which some people do not like. Many thanks for submitting this paper to the special issue of ocean science.

**The Authors thank you for your kind remarks about the paper. Your final revisions have both been implemented into the paper:**
1. **Word has been changed from "astronomic" to "astronomical"**
2. **Figure 6 has been moved to Figure 4 (and subsequently Figures 4 and 5 to Figures 5 and 6).**

[revised manuscript text omitted]